# Coverage Optimization of Heterogeneous Wireless Sensor Network Based on Improved Wild Horse Optimizer

**DOI:** 10.3390/biomimetics8010070

**Published:** 2023-02-06

**Authors:** Chuijie Zeng, Tao Qin, Wei Tan, Chuan Lin, Zhaoqiang Zhu, Jing Yang, Shangwei Yuan

**Affiliations:** 1Electrical Engineering College, Guizhou University, Guiyang 550025, China; 2College of Forestry, Guizhou University, Guiyang 550025, China; 3College of Computer Science and Technology, Guizhou University, Guiyang 550025, China; 4Power China Guizhou Engineering Co., Ltd., Guiyang 550001, China

**Keywords:** heterogeneous wireless sensor network, improved wild horse optimizer, coverage optimization, coverage ratio, connectivity ratio

## Abstract

One of the most important challenges for heterogeneous wireless sensor networks (HWSNs) is adequate network coverage and connectivity. Aiming at this problem, this paper proposes an improved wild horse optimizer algorithm (IWHO). Firstly, the population’s variety is increased by using the SPM chaotic mapping at initialization; secondly, the WHO and Golden Sine Algorithm (Golden-SA) are hybridized to improve the WHO’s accuracy and arrive at faster convergence; Thirdly, the IWHO can escape from a local optimum and broaden the search space by using opposition-based learning and the Cauchy variation strategy. The results indicate that the IWHO has the best capacity for optimization by contrasting the simulation tests with seven algorithms on 23 test functions. Finally, three sets of coverage optimization experiments in different simulated environments are designed to test the effectiveness of this algorithm. The validation results demonstrate that the IWHO can achieve better and more effective sensor connectivity and coverage ratio compared to that of several algorithms. After optimization, the HWSN’s coverage and connectivity ratio attained 98.51% and 20.04%, and after adding obstacles, 97.79% and 17.44%, respectively.

## 1. Introduction

Over the past decade, wireless sensor networks (WSNs) have been used in different fields, such as urban management, environmental monitoring, disaster prevention, and military applications, etc. [1,2,3,4]. A large number of tiny sensors make up the self-organizing distributed network system known as the WSN, and the sensors are typically heterogeneous. In WSN applications, coverage and connectivity are important indicators for determining whether real-time data can be provided to users through the inter-collaboration of sensors. However, the traditional WSN coverage approach deploys sensors at random. This approach will result in insufficient coverage, causing communication conflicts [5,6]. In existing research, scholars usually consider coverage when optimizing HWSN coverage, but connectivity is frequently overlooked. Therefore, this paper studies how to improve the coverage and connectivity of HWSNs.

The swarm intelligence (SI) optimization algorithm is a biologically inspired method that is one of the most successful strategies for solving optimization problems [7,8]. It is characterized by a fast search speed and strong search capability, avoiding complex theoretical derivation. Examples include particle swarm optimization (PSO) [9], bald eagle search optimization algorithm (BES) [10], cuckoo search (CS) [11], sparrow search algorithm (SSA) [12], northern goshawk optimization (NGO) [13], mayfly optimization algorithm (MA) [14], gray wolf optimization algorithm (GWO) [15], Harris hawks optimization (HHO) [16], coot optimization algorithm (COOT) [17], wild horse optimizer (WHO) [18] and other algorithms.

The WHO was put forth by Naruei et al. in 2021 as a method of solving algebraic optimization issues. Its optimization performance has significant advantages over the majority of classical algorithms, and it has been widely used to solve various engineering problems. In 2022, Milovanović et al. applied the WHO to multi-objective energy management in microgrids [19]. Ali et al. applied the WHO to the frequency regulation of a hybrid multi-area power system with a new type of combined fuzzy fractional order PI and TID controllers [20]. Furthermore, many researchers have improved WHO to improve its optimization capability. In 2022, Li et al. proposed a hybrid multi-strategy improved wild horse optimizer, which can improve the algorithm’s convergence speed, accuracy, and stability [21]. Ali et al. proposed an improved wild horse optimization algorithm for reliability-based optimal DG planning of radial distribution networks. This algorithm is a high-performance optimization method in terms of exploration–exploitation balance and convergence speed [22].

In 2017, Tanyildizi et al. proposed the Gold-SA algorithm [23], which is based on the sine trigonometric function. This algorithm uses a golden sine operator to condense the solution space, efficiently avoiding the local optimal outcome and quickly approaching the global optimum. Additionally, the algorithm contains few parameters and algorithm-dependent operators, which can be well integrated with the other algorithms. In 2022, Wang et al. proposed an improved crystal structure algorithm for engineering optimization problems. This algorithm makes good use of the relationship between the golden sine operator and the unit circle to make the algorithm exploration space more comprehensive, which can effectively speed up the convergence rate of the algorithm [24]. Yuan et al. proposed a hybrid golden jackal optimization and golden sine algorithm with dynamic lens imaging learning for global optimization problems; the golden sine algorithm is integrated to improve the ability and efficiency of golden jackal optimization [25]. In 2023, Jia et al. proposed the fusion swarm-intelligence-based decision optimization for energy-efficient train-stopping schemes. Their algorithm incorporates the golden sine strategy to improve the performance of the algorithm [26].

In recent years, SI optimization algorithms have been used by many scholars for the study of WSN coverage optimization, and fruitful results have been achieved with the continuous development of SI. In 2013, Huang et al. proposed an AFSA-based coverage optimization method for WSN. Simulation results show that AFSA increases the sensors’ coverage in WSN [27]. In 2015, Zhang proposed a hybrid algorithm of particle swarm and firefly, with particle swarm as the main body and firefly for local search, thus improving the sensor coverage [28]. In 2016, Wu et al. suggested an improved adaptive PSO-based coverage optimization. This approach first increases the evolution factor and aggregation factor to improve the inertia weights, and then, in order to ensure that the particle population is diverse, it introduces a collision resilience strategy during each iteration of the algorithm [29]. In 2018, Lu et al. proposed an FA-based WSN coverage optimization technique that involves switching out two sensors’ placements at once to increase network coverage [30]. In 2019, Nguyen et al. suggested a powerful genetic algorithm based on coverage optimization, effectively addressing various drawbacks of the current metaheuristic algorithms [31].

Although these SI optimization algorithms have produced many positive results, there is still room for further research into the algorithm’s performance and the optimization of WSN coverage. This research suggests an IWHO to optimize sensor coverage and connectivity. The main contributions are the following:We improve the WHO algorithm in order to achieve better optimization. The SPM chaotic map is used to improve the population’s quality. The WHO and Golden-SA are hybridized to improve the WHO’s accuracy and arrive at faster convergence. The Cauchy variation and opposition-based learning strategies are also used to avoid falling into a local optimum and broaden the search space.We test 23 test functions and compare the results to the performance of the IWHO and seven other algorithms. The findings reveal that IWHO has a stronger optimization performance than the others. We use the IWHO to optimize the coverage of a homogeneous WSN and compare the performance with five other algorithms and four improved algorithms proposed in References. The experimental data demonstrate that IWHO can optimize WSN coverage more effectively.The HWSN coverage problem is optimized using the IWHO, which significantly increases coverage and the connectivity ratio. With the situation of barriers, the same high level of coverage and connectivity of sensors is attained.

## 2. WSN Coverage Model

Suppose that the monitoring area is a two-dimensional region with an area of *M × N*. The *n* sensors are randomly arranged in this area and they can be expressed as *U* = {*u_1_*, *u_2_*…*u_n_*}. Assume that the sensors are heterogeneous with different sensing radii *R_s_* and communication radii *R_c_*, and *R_c_ ≥* 2*R_s_*. Every sensor can move, and their position can be updated instantly. The sensor is centered on itself and has a sensing radius *R_s_* as its radius, covering a circular area. If the coordinates of the detected arbitrary sensor *U_i_* are (*x_i_*,*y_i_*), the coordinates of the target detected sensor *O_j_* are (*x_j_*,*y_j_*). The Euclidean distance from the detected arbitrary sensor *U_i_* to the target detected sensor *O_j_* is expressed as:(1)$$d(U_{i},O_{j})=\sqrt{{\left(x_{i}-x_{j}\right)}^{2}+{\left(y_{i}-y_{j}\right)}^{2}}$$

The probability of sensor *O_i_* being perceived by sensor *U_i_* is denoted by *p*(*U_i_*,*O_i_*). It signifies that the goal is covered and the probability is 1 when the distance between sensors is smaller than *R_s_*. It is 0 if it is not covered. The expression is as follows:(2)$$p\left(U_{i},O_{j}\right)=\left\{\begin{array}{l} \begin{array}{cc} 0 & R_{s}>d\left(U_{i},O_{j}\right) \end{array}\\ \begin{array}{cc} 1 & R_{s}\leq d\left(U_{i},O_{j}\right) \end{array} \end{array}\right.$$

Joint sensors’ perception probabilities are defined as follows:(3)$$P\left(U,O_{j}\right)=1-\prod_{i=1}^{n}\left[1-p\left(U_{i},O_{j}\right)\right]$$

The coverage ratio is an important indicator of the HWSN deployment problem. The coverage ratio is calculated as follows:(4)$$f_{1}=Cov=\frac{\sum_{j=1}^{M\times N}P\left(U,O_{j}\right)}{M\times N}$$

The utilization of sensor coverage is evaluated using coverage efficiency. Higher coverage efficiency means achieving the same coverage area with fewer sensors. It is determined by dividing the region’s effective coverage range by the sum of all of the sensors’ coverage ranges. The coverage efficiency *CE* is calculated as shown in the following equation, where *A_i_* denotes the area covered by the *i*-th sensor:(5)$$CE=\frac{\underset{i=1\ldots n}{\cup }A_{i}}{\sum_{i=1\ldots n}A_{i}}$$

In the coverage problem, the connectivity ratio is equally as important as the coverage among sensors. To ensure the reliability of network connectivity, each sensor should be able to connect with at least two or more sensors. If the separation between two sensors is within the *R_c_*, it is known that the two sensors can be connected to each other. The connectivity ratio between sensor *O_i_* and target detection sensor *O_j_* (*i ≠ j*) can be defined as:(6)$$p\left(O_{i},O_{j}\right)=\left\{\begin{array}{l} \begin{array}{cc} 0 & R_{c}>d\left(O_{i},O_{j}\right) \end{array}\\ \begin{array}{cc} 1 & R_{c}\leq d\left(O_{i},O_{j}\right) \end{array} \end{array}\right.$$

As shown in Figure 1, a network is connected if there is a path between any two sensors. The connectivity ratio is the proportion of connected paths to the maximum connected paths between sensors. The expression is as follows:(7)$$f_{2}=\frac{\sum_{i=1}^{n}\sum_{j=1}^{n}P\left(O_{i},O_{j}\right)}{n(n-1)/2}(i\neq j)$$

The amount of paths between any two nodes is *n*(*n* − 1)/2. Therefore, according to sensor coverage and connection, the objective function is:(8)$$\max F(f_{1},f_{2})=w_{1}f_{1}+w_{2}f_{2}\;st.\left\{\begin{array}{c} w_{1}+w_{2}=1\\ \sum_{j=1}^{M\times N}P(U,O_{j})\leq M\times N\\ p(U_{i},O_{j})\geq 0\\ p(O_{i},O_{j})\geq 0 \end{array}\right.$$

After several experiments, the values of *w*_1_ and *w*_2_, respectively were 0.9 and 0.1.

## 3. Wild Horse Optimizer

The social behavior of wild horses served as a model for the WHO. In the population construction of wild horses, there exist stallions and the rest of the horse herd. The WHO is designed and optimized for various problems based on group behavior, grazing, mating, dominance, and leadership among the stallions and herds in the wild horse population.
Establishing an initial population and choosing leaders

The population members are first distributed at random throughout the search ranges. In the beginning, we group this population. If there are *N* members overall, the number of stallions is *G* = (*N* × *P_S_*). *P_S_* represents the percentage of stallions in the herd, and it serves as the algorithm’s control parameter. The algorithm begins with the group leaders being chosen at random, and as the algorithm progresses, the leaders are chosen based on which group member has the best fitness function.
Grazing and mating of horses

The stallion is regarded as the center of the grazing area, and the group members move about the center to promote grazing behavior. We propose Equation (9) to model the grazing behavior. Members of the group move and conduct searches with varying radii around the leader:(9)$$X_{i,G}^{j}=2Z\cos(2\pi RZ)*(Stallion^{j}-X_{i,G}^{j})+Stallion^{j}$$
where Stallion is the leader’s position and *R* is a random number within [−2, 2]; *Z* is calculated as:(10)$$P=\vec{R_{1}}<TDR;\;IDX=(P==0);\;Z=R_{2}\Theta IDX+\vec{R_{3}}\Theta (\sim IDX)$$
where *P* is a vector consisting of 0 and 1. The random numbers $\vec{R_{1}}$, $R_{2}$, and $\vec{R_{3}}$ have a uniform distribution and fall between [0, 1]. Returns for the *IDX* indices of the random vector $\vec{R_{1}}$ that satisfy the condition (*P* == 0). During algorithm execution, *TDR* declines, starting at 1 and eventually reaching 0. The expression is as follows:(11)$$TDR=1-iter\frac{1}{\max iter}$$
where *iter* indicates how many iterations are being performed right now, and max*iter* indicates the maximum number of iterations.

In order to simulate the behavior of horses leaving and mating, Equation (12) has proposed the same *Crossover* operator as the mean value type:(12)$$X_{G,K}^{p}=Crossover(X_{G,i}^{q},X_{G,j}^{z}),i\neq j\neq k,p=q=end;Crossover=Mean$$

Group is led by a leader

In nature, leaders mostly guide groups to appropriate living environments. If another population dominates the habitat, then that population must leave it. Equation (13) allows calculation of the location of the next habitat searched by the leader in each population:(13)$$Stallion_{Gi}=\left\{\begin{array}{c} 2Z\cos(2\pi RZ)*(WH-Stallion_{Gi})+WH,\begin{array}{cc} if & R_{3}>0.5 \end{array}\\ \;2Z\cos(2\pi RZ)*(WH-Stallion_{Gi})-WH,else \end{array}\right.$$
where *WH* is the current location of the most suitable habitat, and *R*, *RZ*, and *Z* are defined as before.
Exchange of leaders

As mentioned in the population initialization phase, the two positions are switched if a group member has a greater fitness value than the leader:(14)$$Stallion_{Gi}=\left\{\begin{array}{c} X_{Gi},\begin{array}{cc} if & \cos t(X_{Gi})<\cos t(Stallion_{Gi}) \end{array}\\ \;\;Stallion_{Gi},else \end{array}\right.$$

## 4. Improved Wild Horse Optimizer

To address the issues of the original algorithm, including the difficulty of escaping local optima, lack of accuracy during convergence, and slow speed, three methods are introduced in this study as ways to enhance the WHO algorithm. They improve the starting population, boost optimization speed and accuracy, increase the optimal solution’s disruption, and aid the algorithm’s exit from the local optimum.

### 4.1. SPM Chaotic Mapping Initialization Population

The optimization ability can benefit from a diversified initial population. In the original WHO algorithm, the rand function is used to randomly initialize the population, which results in uneven distribution and overlapping individuals, and the population diversity decreases rapidly in the later iterations. Chaos is a unique and widespread form of acyclic motion in nonlinear systems, which is widely used in population intelligence algorithms for optimizing the diversity of populations because of its ergodic and stochastic nature. In this paper, we introduce the SPM chaotic mapping model, which has superior chaotic and ergodic properties [32]. The expression is shown as follows:(15)$$X\left(t+1\right)=\left\{\begin{array}{c} \mathrm{mod}(\frac{x(t)}{\eta }+\mu \sin(\pi x(t))+r,1),0\leq x(t)<\eta \\ \mathrm{mod}(\frac{x(t)/n}{0.5-\eta }+\mu \sin(\pi x(t))+r,1),0\leq x(t)<0.5\\ \mathrm{mod}(\frac{1-x(t)/n}{0.5-\eta }+\mu \sin(\pi (1-x(t)))+r,1),0.5\leq x(t)<1-\eta \\ \mathrm{mod}(\frac{1-x(t)}{\eta }+\mu \sin(\pi (1-x(t)))+r,1),1-\eta \leq x(t)<1 \end{array}\right.$$

Scholars usually choose different chaotic mapping models for optimization of population initialization of population intelligence algorithms. This paper selects the Logistic mapping and Sine mapping with high usage rate, and compares them with SPM mapping under the condition of setting the same initial value and iterating 2000 times. Figure 2 shows the histograms of the three chaotic mappings, where the horizontal coordinate is the chaotic value and the vertical coordinate is the frequency of that chaotic value. The results prove that SPM mapping has better chaos performance and traversal. SPM mapping was therefore chosen to enhance population variety and make the population distribution more uniform. Figure 3 shows the population distribution of different chaotic mappings when the population size is 2000. The Logistic and Sine mappings show many individuals overlapping at the boundary, whereas the SPM mapping has a more uniform distribution.

### 4.2. Golden Sine Algorithm

To fix the disadvantages of the stallion position update method in the WHO algorithm, the Golden-SA algorithm is used in this research. In this paper, the stallion position is updated using the golden sine operator to condense the algorithm’s solution space and enhance the capability of the optimal search. The expression is shown as follows:(16)$$Stallion_{Gi}=Stallion_{Gi}*\left|\sin(r_{1})\right|-r_{2}\sin(r_{1})*\left|x_{1}*WH-x_{2}*Stallion_{Gi}\right|$$
where *r*_1_, *r*_2_ are random numbers between [0, 2π] and [0, π]. In the following iteration, *r*_1_ determines the distance that individual *i* travels and *r*_2_ determines the direction in which individual *i* travels. *x*_1_ and *x*_2_ are the golden partition coefficients. During the iteration, they are utilized to condense the search space and direct the solution to the place that is globally optimal. Its partitioning implementation is shown below:(17)$$x_{1}=a*\left(1-\tau \right)+b*\tau \;x_{2}=a*\tau +b*\left(1-\tau \right)\;\tau =\left(\sqrt{5}-1\right)/2$$
where *a* and *b* are the initial golden mean, and *τ* is the golden mean ratio.

### 4.3. Cauchy Variation and Opposition-Based Learning

The WHO algorithm does not perturb the optimal solution after each iteration, which can keep the solution in a locally optimal state. To address this issue, we employ Cauchy variation and opposition-based learning strategies to perturb the solution. This was inspired by MAO et al., who proposed a chaotic squirrel search algorithm with a mixture of stochastic opposition-based learning and Cauchy variation in 2021 [33].

#### 4.3.1. Cauchy Variation

Gaussian and Cauchy distributions are two similar classical probability density distribution functions, and Gaussian variation has also been used by scholars in algorithm improvements. Figure 4 displays the probability density function curves for both.

In comparison to the Gaussian distribution, as seen in Figure 4, the Cauchy distribution is longer and flatter at both ends. It flattens out as it approaches 0, moves more slowly, and has a smaller peak close to the origin. As a result, Cauchy variance has better perturbation properties than Gaussian variance. As a result, introducing the Cauchy variation strategy can expand the search space and improve the perturbation ability. The expression is as follows:(18)WH=WH∗(1+(1/Max_iter)∗tan(π∗(rand−0.2)))

#### 4.3.2. Opposition-Based Learning

By building the reverse solution from the existing solution, the opposition-based learning strategy can broaden the solution space of algorithms. To determine which solution is preferable, existing solution is contrasted with the reverse solution. The expression for incorporating the opposition-based learning strategy into the WHO algorithm is as follows:(19)$$WH_{back}=ub+rand*(lb-WH)\;WH=WH_{back}+b_{1}*\left(WH-WH_{back}\right)\;b_{1}={\left(\max\_iter-l/\max\_iter\right)}^{l}$$
where *WH_back_* is the reverse solution to the stallion’s optimal position in the *l*th generation, rand is a random matrix of dimension obeying the standard uniform distribution of (0, 1), upper and lower boundaries are denoted by *ub* and *lb*, respectively, and *b*_1_ denotes the information exchange control parameter.

#### 4.3.3. A Dynamic Selection Probability

This paper sets a dynamic selection probability *P*_z_ to choose the strategy to update the stallion position more appropriately. The *P*_z_ is shown as follows:(20)$$P_{z}=-\exp{\left(1-l/\max\_iter\right)}^{20}+0.05$$

*P*_z_ will be compared to a number chosen at random between (0,1). If *P*_z_ > rand then a opposition-based learning strategy starts to work. If *P*_z_ < rand then the Cauchy variation strategy is utilized to disturb at the present stallion.

### 4.4. The Pseudo Code of IWHO

Initialize the first population of horses using the new SPM chaotic sequenceInput IWHO parameters, *P*_C_ = 0.13, *P*_S_ = 0.1, *a* = π, *b* = −πCalculate the fitnessCreate foal groups and select stallionsFind the best horseWhile the end criterion is not satisfiedCalculate *TDR* using Equation (11)For number of stallionsCalculate *Z* using Equation (10)For number of foals of any groupIf rand > *P_C_*Update the position of the foal using Equation (9)ElseUpdate the position of the foal using Equation (12)EndEndUpdate the position of the *StallionG_i_* using Equation (16)If cost (*StallionG_i_*) < cost (Stallion)Stallion = *StallionG_i_*EndSort foals of group by costSelect foal with minimum costIf cost (foal) < cost (Stallion)Exchange foal and Stallion position using Equation (14)EndCalculate *P_z_* using Equation (20)If *P_z_* < randUpdate the position of *Stallion* using Equation (19)ElseUpdate the position of Stallion using Equation (18)EndEndUpdate optimumEnd

### 4.5. Time Complexity Analysis

Time complexity is a significant factor in determining an algorithm’s quality and shows how effectively it operates. The time complexity of the WHO algorithm can be represented as *O*(*N×D×L*), where *N* is the entire population, *D* is the search space’s dimension, and *L* is the maximum number of iterations. The following is a depiction of the IWHO algorithm’s time complexity analysis:The population is initialized using the SPM chaotic mapping model, and the time complexity is as follows:
(21)$$T_{1}=O\left(\left(N_{1}+N_{2}\right)\times D\right)=O\left(N\times D\right)$$
The position update formula of the original stallion is replaced by the golden sine strategy without the addition of any extra strategies. The time complexity is as follows:


(22)
$$T_{2}=O\left(N\times D\times L\right)$$


After iterations, incorporating the Cauchy variance and opposition-based learning, the time complexity is as follows:


(23)
$$T_{3}=O\left(N\times D\times L\right)+O\left(N\times D\times L\right)=O\left(N\times D\times L\right)$$


Thus, the IWHO time complexity is as follows:


(24)
$$T_{1}+T_{2}+T_{3}=O\left(N\times D\right)+O\left(N\times D\times L\right)+O\left(N\times D\times L\right)=O\left(N\times D\times L\right)$$


In conclusion, IWHO has the same time complexity as WHO, and the three improvement techniques do not make the algorithm’s time complexity any more difficult.

## 5. IWHO Algorithm-Based Coverage Optimization Design

The process of finding a suitable habitat for a horse herd is analogous to the process of obtaining the optimal coverage of sensors, and the position of the stallion represents the coordinates covered by the sensors. Using the same number of sensors to cover a bigger area while maintaining effective communication is the aim of WSN optimization coverage based on the IWHO. These are the steps:

Step 1: Enter the size of the area to be detected by the WSN, the number of sensors, sensing radius, communication radius, and the IWHO algorithm’s settings;

Step 2: The population is initialized according to Equation (15), where each individual represents a coverage scheme. At this step, the sensors are dispersed randomly around the monitoring region, and Equation (8) is used to determine the initial coverage and connectivity;

Step 3: Update the location information of stallions and foals, and calculate the corresponding adaptation degree. Update the coverage ratio and connectivity ratio according to Equation (8). Find the optimal sensor location;

Step 4: Create a new solution by perturbing at the optimal solution position through dynamic probabilistic selection of a Cauchy variation or a opposition-based learning strategy;

Step 5: Immediately exit the loop if the condition is met. Output the sensor’s best coverage scheme.

## 6. Simulation Experiments and Analysis

### 6.1. IWHO Algorithm Performance Test Analysis

#### 6.1.1. Simulation Test Environment

The environment for this simulation test was: Windows 10 Professional, 64-bit OS, Intel(R) Core (TM) i5-4210H CPU @2.90 GHz, 8GB. The simulation software was MATLAB 2016a.

#### 6.1.2. Comparison Objects and Parameter Settings

In this paper, the WHO, SSA, NGO, MA, PSO, COOT, GWO and the IWHO algorithms were selected for comparison. In order to make a fair comparison between each algorithm, we have unified the number of consumed fitness evaluations in the experiment, and the number of consumed fitness evaluations by each algorithm is 30,000. The parameters ware set as shown in Table 1.

In GWO, *α* represents the control parameter of a grey wolf when hunting prey. In SSA, ST represents the alarm value, PD represents the number of producers, SD represents the number of sparrows who perceive the danger. In MA, *g* represents the inertia weight, *a*_1_ represents the personal learning coefficient, and *a*_2_ and *a*_3_ represent the global learning coefficient. In PSO, *c*_1_ and *c*_2_ represent the learning coefficient, *w*_min_ and *w*_max_ represent the upper and lower limits of inertia weight. In COOT, *R*, *R*_1_ and *R*_2_ are random vectors along the dimensions of the problem.

#### 6.1.3. Benchmark Functions Test

To verify the IWHO algorithm’s capacity for optimization, 23 benchmark functions were chosen for simulation in this study. There are three categories of test function, among which F1–F7 listed in Table 2 are single-peak test functions, F8–F13 are multi-peak test functions listed in Table 3 and F14–F23 are fixed-dimension test functions listed in Table 4. The single-peak test functions are characterized by a single extreme value and are used to test the convergence speed and convergence accuracy of the IWHO. The multi-peak test functions are characterized by multiple local extremes, which can be applied to evaluate the IWHO algorithm’s local and global search capabilities.

#### 6.1.4. Simulation Test Results

After 30 independent runs, Table 5 displays the average value and standard deviation for eight algorithms. The IWHO outperforms others. For the single-peak test functions F1–F4 and F7, the IWHO has the best performance and the fastest convergence speed, with a standard deviation of 0, demonstrating that it is robust and stable. IWHO has the second-best performance for F5 and F6. For F9 and F11, although most of the algorithms converge to the ideal optimum, as illustrated in Figure 5i,k, the IWHO converges faster. Both the IWHO and SSA algorithms have the best results for F10, however, the IWHO goes through fewer iterations, as can be seen in Figure 5j. For F13, both the PSO algorithm and the SSA algorithm outperform the IWHO, which is ranked third among the eight optimization algorithms. The results in Table 5 and Figure 5 for the fixed-dimension test functions show that the IWHO converges to the theoretical optimum with a standard deviation of 0.

In conclusion, the results of 30 independent experiments with eight different algorithms and 23 benchmark test functions show that the IWHO can converge to the ideal optimal values for 14 test functions, F1, F3, F9, F11, F14–F23. This means that the convergence accuracy of IWHO has been improved a lot, but when optimizing some test functions, although the optimal value can be obtained, the convergence speed can still be improved. Additionally, some test functions still fall into local optimization. Among the 23 test results, IWHO ranked first 20 times, second twice, and third once. It can be concluded that the IWHO performs better than the WHO algorithm and effectively improves some of its flaws. Similarly, the IWHO has significant advantages over other algorithms.

#### 6.1.5. Wilcoxon Rank Sum Test

Such data analysis lacks integrity and scientific validity if it only compares and analyzes the mean and standard deviation of the different algorithms. We therefore chose a non-parametric statistical test, the Wilcoxon rank sum test, to further validate the algorithm’s performance. We ran each algorithm independently for 30 times in 23 test functions. For Wilcoxon rank sum test and *p* calculation, we compared the experimental results of the other algorithms with those of the IWHO. When *p* < 5%, it was marked it as “+”, indicating that IWHO was better than the comparison algorithm. When *p* > 5%, it was marked as “−”, indicating that IWHO is inferior to the comparison algorithm. When *p* was equal to 1, it indicates that it is not suitable for judgment.

The comparison results are shown in Table 6. In the comparison of various algorithms, most of the rank sum test *p* values are less than 0.05, which indicates that the IWHO has significant differences with other algorithms; that is, the IWHO algorithm has better optimization performance.

### 6.2. Coverage Performance Simulation Test Analysis

To verify the performance of IWHO in optimizing coverage in HWSNs, three experiments were set up by simulating different scenarios and setting different parameters.
Experiment 1 used homogenous sensors, and the IWHO was used to improve the coverage ratio of WSN. In order to demonstrate the IWHO algorithm’s efficacy, five algorithms and four4 improved algorithms were chosen for comparison.The results of Experiment 1 show how effective the IWHO is in solving the sensor coverage optimization issue. In Experiment 2, the IWHO was used in an HWSN to increase sensor coverage and ensure sensor connectivity.In Experiment 3, an obstacle was added to the monitoring area to simulate a more realistic scenario. The sensors must avoid the obstacle for coverage, and the IWHO was used to improve the coverage and connectivity ratio of the sensors in the monitoring area.

#### 6.2.1. Simulation Experiment 1 Comparison Results Analysis

In Experiment 1, we only considered the improvement in coverage ratio using Equation (4) as the objective function. To avoid the possibility of the algorithm, the algorithm was run thirty times independently. Table 7 displays the settings for the sensor parameters.

Figure 6a shows how the sensors were initially covered in the monitoring area at random. With the optimization of the IWHO, the overlapping sensors started to decrease. Finally, as shown in Figure 6b, they were evenly distributed in the monitoring area. Table 8 demonstrates that the initial coverage reached 79.13%, while after the optimization of the IWHO the coverage reached 97.58%, which is an improvement of 18.45%. At the beginning, there were more redundant sensors in the region, and the region had more obvious energy voids and appeared to be cluttered. After the optimization of the algorithm, the sensor distribution became obviously uniform and the coverage ratio was significantly improved. As a result, the IWHO is effective for WSN coverage optimization.

WHO, PSO, WOA, BES, HHO algorithms were selected for further comparative experiments. All algorithms were run independently thirty times. Table 6 displays the settings for the sensor parameters.

Table 9 demonstrates that the IWHO, with a coverage ratio of 97.58%, has the best optimization result. The results of the experiments can prove that the IWHO shows better results than the WHO algorithm in solving the WSN coverage optimization problem, with an average coverage improvement of 5.52% and a coverage efficiency improvement of 3.93% after thirty independent runs. It also outperforms the other four algorithms. The coverage ratio after IWHO optimization is 14.56% higher than the worst PSO algorithm and 1.1% higher than the best HHO algorithm. Figure 7 shows that the IWHO maintains a better coverage ratio for almost all iterations. Additionally, the IWHO algorithm’s coverage efficiency is the highest, coming in at 69.08%, surpassing the PSO algorithm’s coverage efficiency by 10.3%. It demonstrates that the IWHO is optimized by reducing redundancy in sensor coverage.

To further verify the superiority of IWHO in optimizing sensors coverage, the IWHO was compared with the four improved algorithms. To ensure fairness, the parameter settings are kept consistent with those in reference.

As Table 10, Table 11, Table 12 and Table 13 demonstrate, with the same parameter values, the IWHO improves WSN coverage.

#### 6.2.2. Simulation Experiment 2 Comparison Results Analysis

In a complex sensor coverage environment, it is difficult to unify the types of sensors. Therefore, more of the HWSN is often covered in real environments. In Experiment 2, two different types of sensors were set up, and the sensors were dispersed at random throughout the monitoring area. The IWHO was used to optimize the HWSN coverage. Table 14 displays the settings for the sensor parameters. *N*_1,2_ indicates the number of two types of sensors.

As shown in Figure 8a,b and Table 15, IWHO improved coverage equally well when two kinds of different sensors are deployed. The optimized coverage achieved 98.51%, a 17.08% improvement over the initial state. In contrast to Experiment 1, IWHO improved connectivity while improving coverage. As illustrated in Figure 8c,d and Table 16, some of the sensors were not connected in the initial state, and the connectivity ratio was only 16.03%. However, after optimization, the connectivity ratio rose to 20.04%, and the overall network connectivity improved.

#### 6.2.3. Simulation Experiment 3 Comparison Results Analysis

In the actual coverage area, there are some non-deployable units. In order to simulate a more realistic simulation scenario, a 25 m × 25 m obstacle was added to the monitoring area. Experiment 3 evaluated the IWHO algorithm’s performance in optimizing HWNs coverage with the existence of obstacles. Table 13 displays the settings for the sensor parameters.

Similarly, as shown in Figure 9a,b and Table 17, IWHO can also effectively improve the coverage ratio after adding obstacles to the monitoring area, reaching 99.25%, which is 16.96% higher than the initial state. As shown in Figure 9c,d and Table 18, some sensors were not connected at the start. Following optimization, the network connectivity ratio increased to 17.44%.

## 7. Conclusions

This paper proposes an IWHO to increase the coverage and connectivity of HWSNs. The WHO algorithm’s flaws include a slow rate of convergence, poor convergence accuracy, and a propensity for falling into a local optimum. How can the algorithm be made to perform better as a result? The population is initialized using the SPM chaotic mapping model, followed by the integration of the Golden-SA to enhance the algorithm’s search for optimization, and finally, the best solution is perturbed using the opposition-based learning and Cauchy variation strategy to avoid a local optimum being reached. This study evaluates the optimization performance of the IWHO, the WHO algorithm, and another seven algorithms using 23 benchmark test functions to confirm the performance of the IWHO. According to the results of the simulation, the IWHO outperforms other algorithms in terms of convergence accuracy and speed, as well as the ability to escape local optimization. Although the IWHO can find the ideal optimal value for the majority of test functions, the convergence rate can still be improved. For individual test functions, IWHO still falls into local optimization. To further verify IWHO’s superiority in optimizing sensor coverage, five other algorithms and four improved algorithms were used. The outcomes demonstrate that the IWHO achieves optimum sensor coverage and is superior to other algorithms. The coverage issue of HWSNs is finally resolved via IWHO, which optimizes sensor coverage and enhances sensor connectivity. After adding an obstacle, it can also be optimized to obtain good results.

However, the current research is still insufficient. We can continue enhancing the IWHO to increase its performance and researching the IWHO for multi-objective optimization so that it can handle more complex HWSN coverage and engineering issues.

## Figures and Tables

**Figure 1 biomimetics-08-00070-f001:**
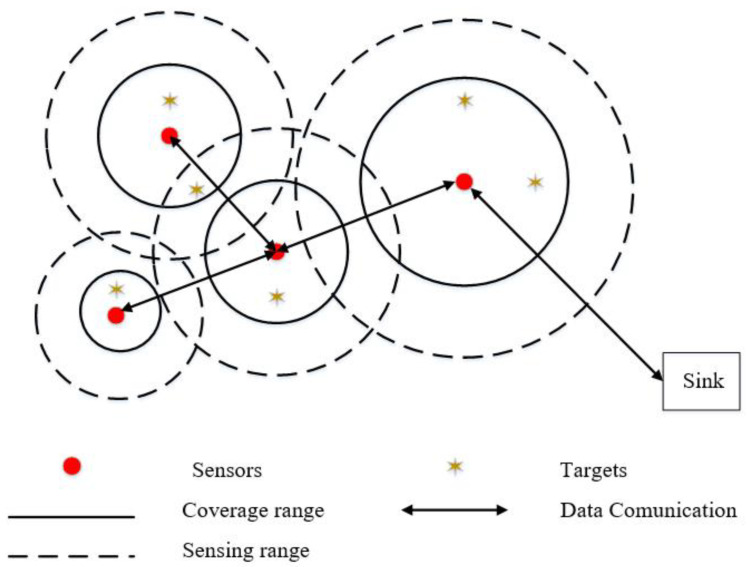
HWSN coverage model.

**Figure 2 biomimetics-08-00070-f002:**
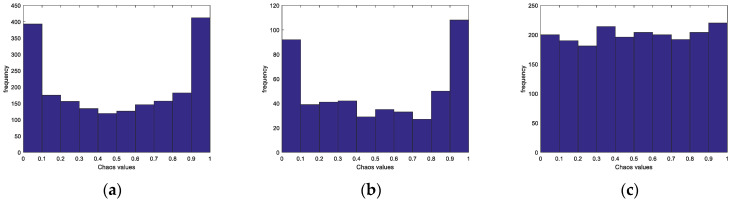
Chaotic Mapping Histogram. (**a**) Logistic mapping; (**b**) Sine mapping; (**c**) SPM mapping.

**Figure 3 biomimetics-08-00070-f003:**
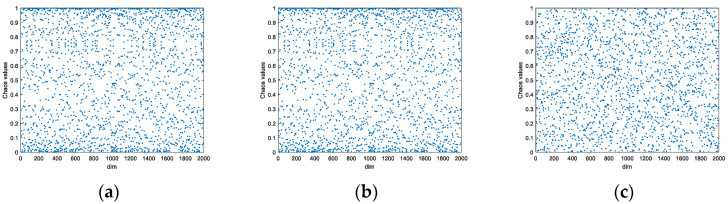
Chaotic Mapping Scatter Diagram. (**a**) Logistic mapping; (**b**) Sine mapping; (**c**) SPM mapping.

**Figure 4 biomimetics-08-00070-f004:**
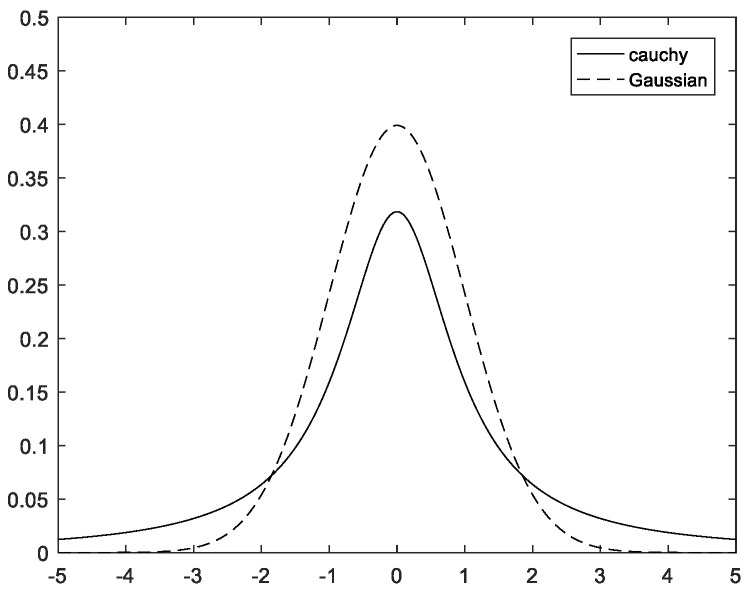
Curves of the probability density functions.

**Figure 5 biomimetics-08-00070-f005:**
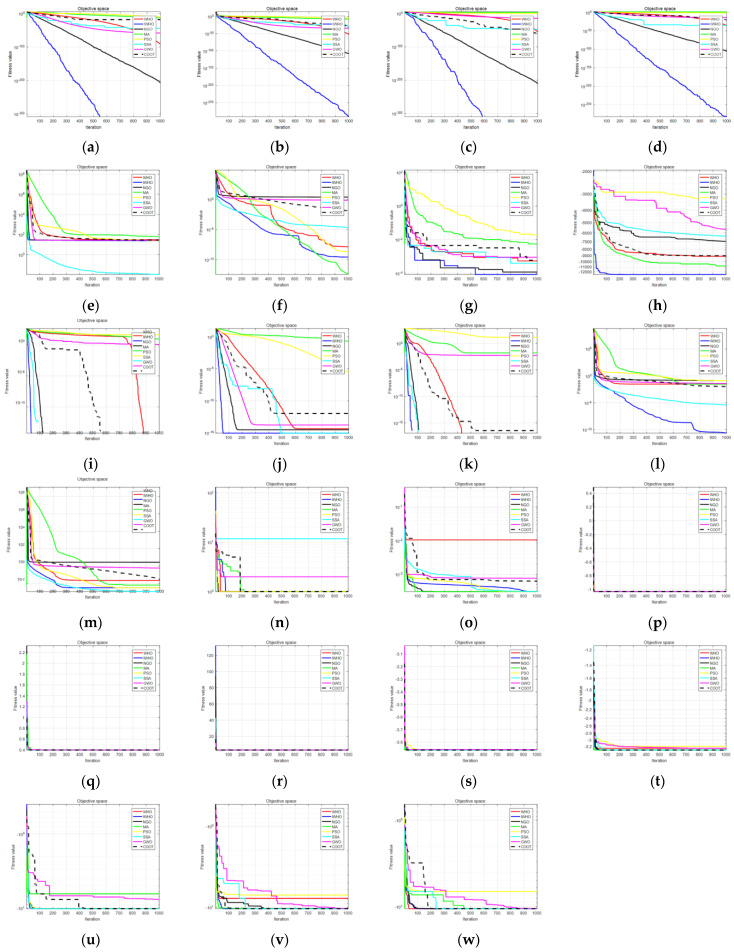
Comparison of convergence curves. (**a**) F1; (**b**) F2; (**c**) F3; (**d**) F4; (**e**) F5; (**f**) F6; (**g**) F7; (**h**) F8; (**i**) F9; (**j**) F10; (**k**) F11; (**l**) F12; (**m**) F13; (**n**) F14; (**o**) F15; (**p**) F16; (**q**) F17; (**r**) F18; (**s**) F19; (**t**) F20; (**u**) F21; (**v**) F22; (**w**) F23.

**Figure 6 biomimetics-08-00070-f006:**
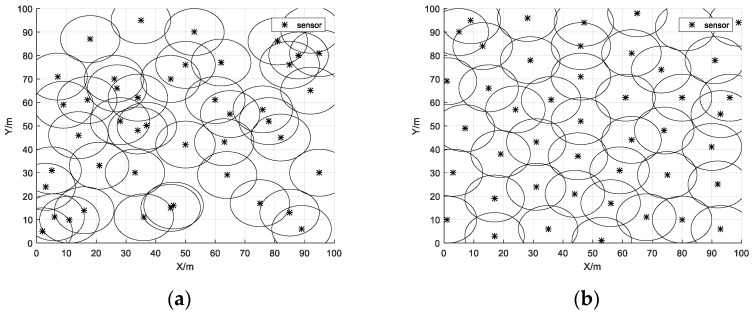
Coverage maps of sensors. (**a**) Initial coverage map of sensors; (**b**) Optimized coverage map of sensors.

**Figure 7 biomimetics-08-00070-f007:**
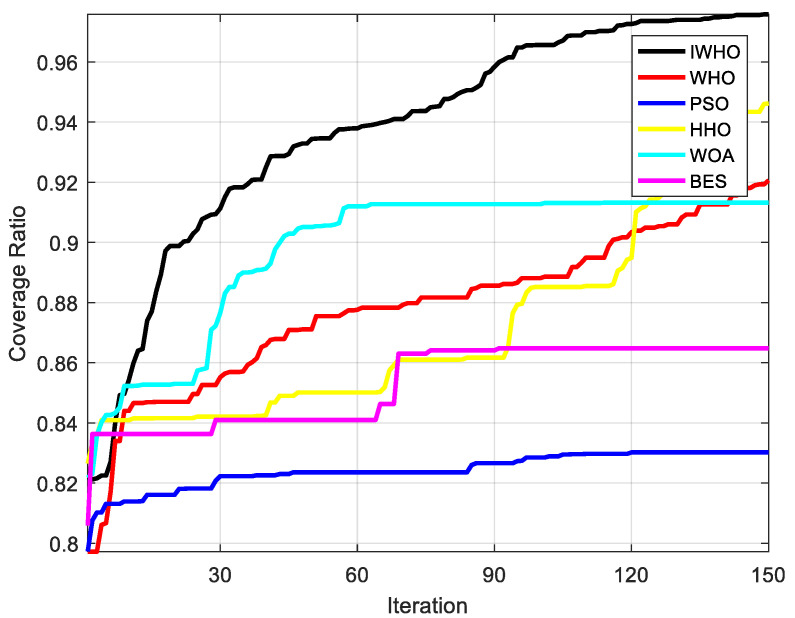
Comparison of coverage convergence curves.

**Figure 8 biomimetics-08-00070-f008:**
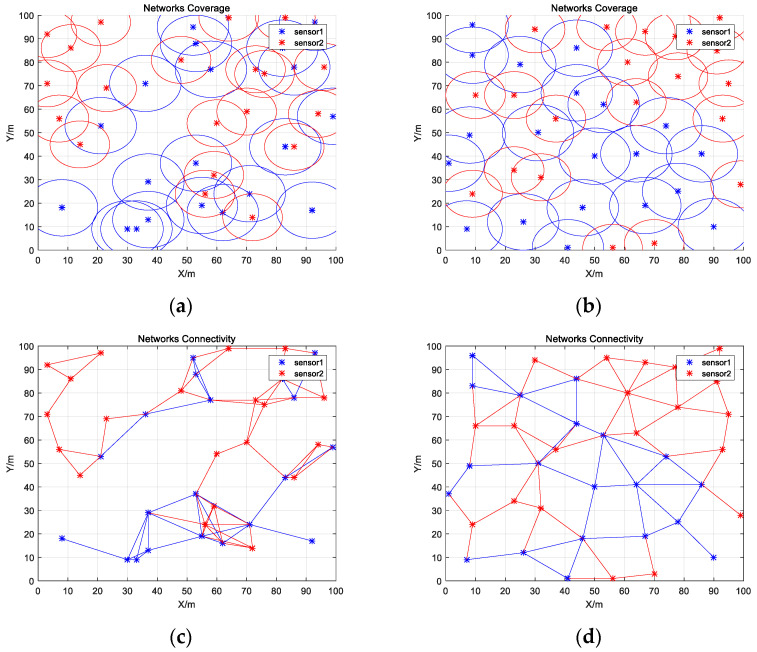
Coverage and connectivity map of sensors. (**a**) Initial coverage map of sensors; (**b**) Optimized coverage map of sensors; (**c**) Initial connectivity map of sensors; (**d**) Optimized connectivity map of sensors.

**Figure 9 biomimetics-08-00070-f009:**
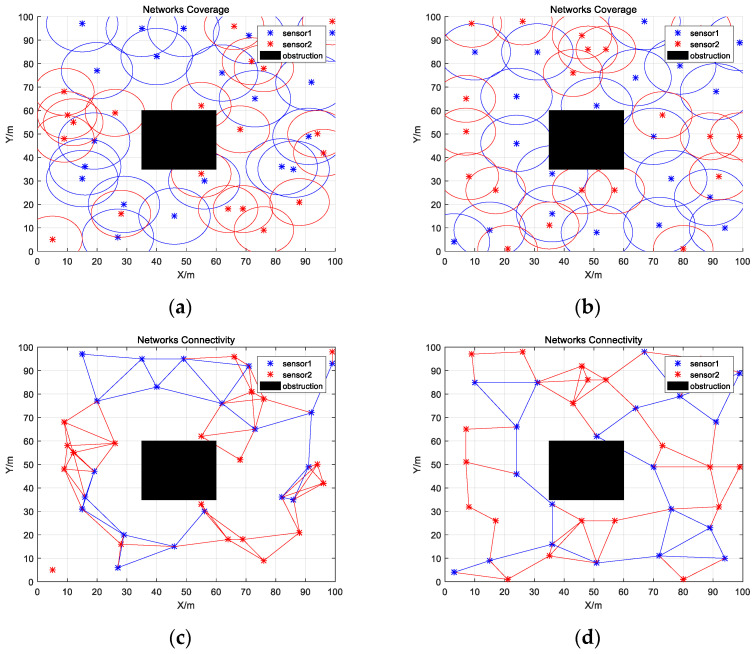
Obstacle coverage and connectivity map of sensors. (**a**) Initial obstacle coverage map of sensors; (**b**) Optimized obstacle coverage map of sensors; (**c**) Initial obstacle connectivity map of sensors; (**d**) Optimized obstacle connectivity map of sensors.

**Table 1 biomimetics-08-00070-t001:** Parameter settings of the algorithm.

Algorithm	Parameters
GWO	*α* = [0, 2]
SSA	*ST* = 0.6, *PD* = 0.7, *SD* = 0.2
MA	*g* = 0.8, *a*_1_ = 1, *a*_2_ = *a*_3_ = 1.5
PSO	*c*_1_*_,_c*_2_ = 2, *w*_min_ = 0.2*, w*_max_ = 0.9
COOT	*R* = [−1, 1], *R*_1_ = *R*_2_ = [0, 1]
WHO	*P_S_* = 0.1, *P_C_* = 0.13
IWHO	*P_S_* = 0.1, *P_C_* = 0.13, *a* = π, *b* = −π

**Table 2 biomimetics-08-00070-t002:** Single-peak test functions.

F	Function	Range	Dim	f_min_
F1	$f_{1}\left(x\right)=\sum_{i=1}^{n}x_{i}^{2}$	[−100, 100]	30	0
F2	$f_{2}\left(x\right)=\sum_{i=1}^{n}\left|x_{i}\right|+\prod_{i=1}^{n}\left|x_{i}\right|$	[−10, 10]	30	0
F3	$f_{3}\left(x\right)={\sum_{i=1}^{n}\left(\sum_{j=1}^{n}x_{j}\right)}^{2}$	[−100, 100]	30	0
F4	$f_{4}\left(x\right)=\max\left\{\left|x_{i}\right|,1\leq i\leq n\right\}$	[−100, 100]	30	0
F5	$f_{5}\left(x\right)=\sum_{i=1}^{n-1}\left[100{\left(x_{i+1}-x_{i}^{2}\right)}^{2}+{\left(x_{i}-1\right)}^{2}\right]$	[−30, 30]	30	0
F6	$f_{6}\left(x\right)={\sum_{i=1}^{n-1}\left(\left[x_{i}+0.5\right]\right)}^{2}$	[−100, 100]	30	0
F7	$f_{7}\left(x\right)=\max\left\{\left|x_{i}\right|,1\leq i\leq n\right\}$	[−1.28, 1.28]	30	0

**Table 3 biomimetics-08-00070-t003:** Multi-peak test functions.

F	Function	Range	Dim	f_min_
F8	$f_{8}\left(x\right)=\sum_{i=1}^{n}-x_{i}\sin(\sqrt{\left|x_{i}\right|})$	[−500, 500]	30	−418.9829 × *n*
F9	$f_{9}\left(x\right)=\sum_{i=1}^{n}x_{i}^{2}-10\;\cos(2\pi x_{i})+10$	[−5.12, 5.12]	30	0
F10	$\begin{array}{l} f_{10}\left(x\right)=-20\exp(-0.2\sqrt{\frac{1}{n}\sum_{i=1}^{n}x_{i}^{2}})-\\ \exp(\frac{1}{n}\sum_{i=1}^{n}\cos(2\pi x_{i}))+20+e \end{array}$	[−32, 32]	30	0
F11	$f_{11}\left(x\right)=\sum_{i=1}^{n}\frac{x_{i}^{2}}{4000}-\prod_{i=1}^{n}\cos(\frac{x_{i}}{\sqrt{i}})-1$	[−600, 600]	30	0
F12	$\begin{array}{l} f_{12}\left(x\right)=\frac{\pi }{n}\left\{\begin{array}{c} 10\sin^{2}(\pi y_{i})+\sum_{i=1}^{n-1}{\left(y_{i}-1\right)}^{2}\\ {}[1+10\sin^{2}(\pi y_{i+1})]+{\left(y_{n}-1\right)}^{2} \end{array}\right\}+\\ \sum_{i=1}^{n}u(x_{i},10,100,4) \end{array}$	[−50, 50]	30	0
F13	$\begin{array}{l} f_{13}\left(x\right)=0.1\left\{\begin{array}{c} \sin^{2}(3\pi x_{i})+\sum_{i=1}^{n}{\left(x_{i}-1\right)}^{2}\\ \left[1+\sin^{2}(3\pi x_{i}+1)\right]+\\ {\left(x_{n}-1\right)}^{2}\left[1+\sin^{2}(2\pi x_{n})\right] \end{array}\right\}\\ +\sum_{i=1}^{n}u(x_{i},5,100,4) \end{array}$	[−50, 50]	30	0

**Table 4 biomimetics-08-00070-t004:** Fixed-dimension test functions.

F	Function	Range	Dim	f_min_
F14	$f_{14}\left(x\right)={\left(\frac{1}{500}+\sum_{j=1}^{25}\frac{1}{j+\sum_{i=1}^{2}{\left(x_{i}-a_{ij}\right)}^{6}}\right)}^{-1}$	[−65, 65]	2	1
F15	$f_{15}\left(x\right)={\sum_{i=1}^{11}\left[a_{i}-\frac{x_{1}(b_{i}^{2}+b_{i}x_{2})}{b_{i}^{2}+b_{i}x_{3}+x_{4}}\right]}^{2}$	[−5, 5]	4	0.00030
F16	$f_{16}\left(x\right)=4x_{{}_{1}}^{2}-2.1x_{{}_{1}}^{4}+\frac{1}{3}x_{{}_{1}}^{6}+x_{1}x_{2}-4x_{{}_{2}}^{2}+4x_{2}^{4}$	[−5, 5]	2	−1.0316
F17	$f_{17}\left(x\right)={\left(x_{2}-\frac{5.1}{4\pi^{2}}x_{1}^{2}+\frac{5}{\pi }x_{1}-6\right)}^{2}+10(1-\frac{1}{8\pi })\cos x_{1}+10$	[−5, 5]	2	0.398
F18	$\begin{array}{l} f_{18}\left(x\right)=[1+{\left(x_{1}+x_{2}+1\right)}^{2}(19-14x_{1}+3x_{1}^{2}-14x_{2}+6x_{1}x_{2}+3x_{2}^{2})]\\ {}*[30+{\left(2x_{1}-3x_{2}\right)}^{2}*(18-32x_{1}+12x_{1}^{2}+48x_{2}-36x_{1}x_{2}+27x_{2}^{2})] \end{array}$	[−2, 2]	2	3
F19	$f_{19}\left(x\right)=-\sum_{i=1}^{4}c_{i}\exp(-{\sum_{j=1}^{3}a_{ij}(x_{i}-p_{ij})}^{2})$	[1, 3]	3	−3.86
F20	$f_{20}\left(x\right)=-\sum_{i=1}^{4}c_{i}\exp(-{\sum_{j=1}^{6}a_{ij}(x_{i}-p_{ij})}^{2})$	[0, 1]	6	−3.32
F21	$f_{21}\left(x\right)=-{\sum_{i=1}^{5}\left[(X-a_{i}){\left(X-a_{i}\right)}^{T}+c_{i}\right]}^{-1}$	[0, 10]	4	−10.1532
F22	$f_{22}\left(x\right)=-{\sum_{i=1}^{7}\left[(X-a_{i}){\left(X-a_{i}\right)}^{T}+c_{i}\right]}^{-1}$	[0, 10]	4	−10.4028
F23	$f_{23}\left(x\right)=-{\sum_{i=1}^{10}\left[(X-a_{i}){\left(X-a_{i}\right)}^{T}+c_{i}\right]}^{-1}$	[0, 10]	4	−10.5363

**Table 5 biomimetics-08-00070-t005:** Results of single-peak and multi-peak benchmark functions.

Function	Criteria	IWHO	WHO	MA	PSO	NGO	SSA	GWO	COOT
F1	avg	0	6.629 × 10^−91^	4.714 × 10^−12^	2.737 × 10^−10^	3.979 × 10^−211^	1.122 × 10^−43^	4.529 × 10^−59^	1.321 × 10^−39^
	std	0	3.567 × 10^−90^	2.421 × 10^−11^	1.086 × 10^−9^	0	4.747 × 10^−43^	9.959 × 10^−12^	7.114 × 10^−39^
F2	avg	1.025 × 10^−291^	2.506 × 10^−50^	9.438 × 10^−8^	4.021 × 10^−4^	4.77 × 10^−106^	1.461 × 10^−26^	1.109 × 10^−34^	3.414 × 10^−26^
	std	0	1.344 × 10^−49^	1.563 × 10^−7^	3.329 × 10^−3^	2.553 × 10^−105^	6.542 × 10^−26^	1.475 × 10^−34^	1.834 × 10^−25^
F3	avg	0	1.572 × 10^−56^	1.859 × 10^3^	1.085	7.325 × 10^−207^	7.957 × 10^−38^	3.953 × 10^−15^	5.870 × 10^−44^
	std	0	7.437 × 10^−56^	9.828 × 10^2^	3.62 × 10^−1^	0	4.285 × 10^−37^	1.486 × 10^−14^	3.109 × 10^−43^
F4	avg	2.494 × 10^−82^	6.154 × 10^−37^	3.457 × 10^1^	1.102 × 10^1^	4.512 × 10^−103^	1.243 × 10^−37^	1.642 × 10^−14^	3.053 × 10^−7^
	std	0	1.232 × 10^−36^	8.807	3.057 × 10^2^	2.311 × 10^−102^	6.698 × 10^−37^	2.238 × 10^−14^	1.644 × 10^−6^
F5	avg	2.485 × 10^−1^	5.292 × 10^1^	5.338 × 10^1^	2.987 × 10^1^	2.810 × 10^1^	8.877 × 10^−3^	2.686 × 10^1^	2.971 × 10^1^
	std	2.283 × 10^−1^	10.059 × 10^2^	4.787 × 10^1^	1.551 × 10^1^	9.521 × 10^−1^	7.998 × 10^−3^	9.602 × 10^−1^	7.874
F6	avg	4.562 × 10^−10^	1.595 × 10^−8^	6.424 × 10^−14^	8.325 × 10^−10^	2.553	2.641 × 10^−5^	6.713 × 10^−1^	8.996 × 10^−3^
	std	1.356 × 10^−9^	4.095 × 10^−8^	1.867 × 10^−13^	1.826 × 10^−9^	5.301 × 10^−1^	1.352 × 10^−5^	3.432 × 10^−1^	5.114 × 10^−3^
F7	avg	3.703 × 10^−5^	3.498 × 10^−4^	1.234 × 10^−2^	1.982 × 10^−2^	1.079 × 10^−4^	1.077 × 10^−3^	9.155 × 10^−4^	1.098 × 10^−3^
	std	3.150 × 10^−5^	1.879 × 10^−4^	2.861 × 10^−3^	3.175 × 10^−3^	2.009 × 10^−5^	1.331 × 10^−4^	1.368 × 10^−4^	9.571 × 10^−4^
F8	avg	−1.232 × 10^4^	−9.025 × 10^3^	−1.053 × 10^4^	−2.752 × 10^3^	−7.968 × 10^3^	−8.195 × 10^3^	−6.279 × 10^3^	−7.543 × 10^3^
	std	4.721 × 10^2^	7.426 × 10^2^	3.316 × 10^2^	3.766 × 10^2^	5.581 × 10^2^	2.483 × 10^3^	7.877 × 10^2^	6.622 × 10^2^
F9	avg	0	0	7.617	4.905 × 10^1^	0	0	1.518 × 10^−1^	4.661 × 10^−13^
	std	0	0	4.252	1.301 × 10^1^	0	0	8.176 × 10^−1^	2.334 × 10^−12^
F10	avg	8.881 × 10^−16^	3.611 × 10^−15^	3.965 × 10^−1^	5.111 × 10^−6^	3.73 × 10^−15^	8.881 × 10^−16^	1.581 × 10^−14^	2.836 × 10^−11^
	std	0	1.760 × 10^−15^	5.369 × 10^−1^	7.288 × 10^−6^	1.421 × 10^−15^	0	2.495 × 10^−15^	1.525 × 10^−10^
F11	avg	0	0	1.709 × 10^−2^	1.397 × 10^1^	0	0	3.712 × 10^−3^	3.33 × 10^−17^
	std	0	0	1.993 × 10^−2^	3.651	0	0	8.541 × 10^−3^	9.132 × 10^−17^
F12	avg	2.310 × 10^−11^	1.727 × 10^−2^	2.839 × 10^−2^	2.901 × 10^−1^	1.524 × 10^−1^	3.643 × 10^−6^	3.721 × 10^−2^	2.676 × 10^−2^
	std	6.685 × 10^−11^	3.863 × 10^−2^	5.284 × 10^−2^	3.890 × 10^−1^	6.048 × 10^−2^	1.858 × 10^−6^	1.853 × 10^−2^	7.688 × 10^−2^
F13	avg	2.080 × 10^−2^	6.233 × 10^−2^	1.054 × 10^−2^	1.831 × 10^−3^	2.579	5.259 × 10^−3^	6.249 × 10^−1^	5.709 × 10^−2^
	std	3.187 × 10^−2^	3.187 × 10^−2^	2.092 × 10^−2^	4.094 × 10^−3^	4.391 × 10^−1^	6.224 × 10^−3^	2.481 × 10^−1^	4.912 × 10^−2^
F14	avg	9.98 × 10^−1^	9.98 × 10^−1^	9.98 × 10^−1^	9.98 × 10^−1^	9.98 × 10^−1^	1.171 × 10^1^	1.99	9.98 × 10^−1^
	std	0	0	0	3.047 × 10^−12^	0	9.536 × 10^−1^	9.920 × 10^−1^	0
F15	avg	3.075 × 10^−4^	7.653 × 10^−4^	3.075 × 10^−4^	6.441 × 10^−4^	3.075 × 10^−4^	3.163 × 10^−4^	2.036 × 10^−2^	7.271 × 10^−4^
	std	1.038 × 10^−8^	2.375 × 10^−4^	3.833 × 10^−4^	3.369 × 10^−4^	4.578 × 10^−4^	6.587 × 10^−4^	2.491 × 10^−8^	5.714 × 10^−5^
F16	avg	−1.031	−1.031	−1.031	−1.031	−1.031	−1.031	−1.031	−1.031
	std	0	1.570 × 10^−16^	0	0	1.57 × 10^−16^	1.087 × 10^−9^	4.158 × 10^−10^	1.776 × 10^−15^
F17	avg	3.978 × 10^−1^	3.978 × 10^−1^	3.978 × 10^−1^	3.978 × 10^−1^	3.978 × 10^−1^	3.978 × 10^−1^	3.978 × 10^−1^	3.978 × 10^−1^
	std	0	0	1.153 × 10^−7^	0	0	1.180 × 10^−9^	1.498 × 10^−7^	1.667 × 10^−10^
F18	avg	3	3	3	3	3	3	3	3
	std	0	0	3.140 × 10^−16^	7.021 × 10^−16^	0	5.518 × 10^−8^	1.017 × 10^−5^	8.038 × 10^−14^
F19	avg	−3.862	−3.862	−3.862	−3.862	−3.862	−3.862	−3.862	−3.862
	std	0	0	0	0	4.440 × 10^−16^	3.859 × 10^−7^	3.810 × 10^−3^	4.440 × 10^−16^
F20	avg	−3.322	−3.626	−3.322	−3.203	−3.322	−3.322	−3.261	−3.322
	std	0	5.944 × 10^−2^	0	0	5.916 × 10^−7^	1.097 × 10^−6^	6.038 × 10^−2^	1.416 × 10^−10^
F21	avg	−1.015 × 10^1^	6.418	−6.391	−1.015 × 10^1^	−1.015 × 10^1^	−1.015 × 10^1^	−7.604	−1.015 × 10^1^
	std	0	3.735	3.761	0	6.21 × 10^−5^	6.568 × 10^−6^	2.548	1.433 × 10^−10^
F22	avg	−1.040 × 10^1^	−7.765	−1.040 × 10^1^	−7.063	−1.040 × 10^1^	−1.040 × 10^1^	−1.040 × 10^1^	−1.040 × 10^1^
	std	0	2.637	0	3.339	2.660 × 10^−5^	1.666 × 10^−5^	3.949 × 10^−4^	1.810 × 10^−11^
F23	avg	−1.053 × 10^1^	−1.053 × 10^1^	−1.053 × 10^1^	−6.671	−1.053 × 10^1^	−1.053 × 10^1^	−1.053 × 10^1^	−1.053 × 10^1^
	std	0	0	0	3.864	2.882 × 10^−8^	6.094 × 10^−6^	1.722 × 10^−5^	6.436 × 10^−12^

**Table 6 biomimetics-08-00070-t006:** The result of Wilcoxon rank sum test.

Function	WHO	MA	PSO	NGO	SSA	GWO	COOT
F1	1.734 × 10^−6^	1.734 × 10^−6^	1.821 × 10^−6^	1.734 × 10^−6^	1.734 × 10^−6^	1.734 × 10^−6^	1.734 × 10^−6^
F2	1.734 × 10^−6^	1.734 × 10^−6^	7.691 × 10^−6^	1.734 × 10^−6^	1.734 × 10^−6^	1.734 × 10^−6^	1.734 × 10^−6^
F3	1.734 × 10^−6^	1.734 × 10^−6^	4.01 × 10^−5^	1.734 × 10^−6^	1.734 × 10^−6^	1.734 × 10^−6^	1.734 × 10^−6^
F4	1.734 × 10^−6^	1.734 × 10^−6^	1.360 × 10^−5^	1.734 × 10^−6^	1.734 × 10^−6^	1.734 × 10^−6^	1.734 × 10^−6^
F5	3.882 × 10^−6^	6.564 × 10^−6^	2.224 × 10^−6^	1.734 × 10^−6^	1.734 × 10^−6^	1.734 × 10^−6^	5.171 × 10^−1^
F6	1.639 × 10^−5^	1.356 × 10^−1^	1.734 × 10^−6^	1.734 × 10^−6^	1.734 × 10^−6^	1.734 × 10^−6^	1.734 × 10^−6^
F7	1.734 × 10^−6^	1.734 × 10^−6^	1.734 × 10^−6^	1.734 × 10^−6^	1.734 × 10^−6^	1.734 × 10^−6^	1.734 × 10^−6^
F8	1.734 × 10^−6^	2.353 × 10^−6^	1.734 × 10^−6^	1.734 × 10^−6^	4.729 × 10^−6^	1.734 × 10^−6^	1.734 × 10^−6^
F9	1	1.734 × 10^−6^	1.734 × 10^−6^	1	1	2.441 × 10^−4^	1.25 × 10^−2^
F10	7.744 × 10^−6^	1.734 × 10^−6^	1.734 × 10^−6^	2.727 × 10^−6^	1	6.932 × 10^−7^	1.789 × 10^−5^
F11	1	1.734 × 10^−6^	1.734 × 10^−6^	1	1	2.5 × 10^−1^	2.5 × 10^−1^
F12	8.466 × 10^−6^	2.182 × 10^−2^	4.729 × 10^−6^	1.734 × 10^−6^	1.734 × 10^−6^	1.734 × 10^−6^	1.734 × 10^−6^
F13	7.035 × 10^−1^	2.623 × 10^−1^	1.025 × 10^−5^	1.734 × 10^−6^	2.585 × 10^−3^	1.734 × 10^−6^	1.020 × 10^−2^
F14	8.134 × 10^−1^	1.562 × 10^−2^	7.744 × 10^−1^	1.562 × 10^−2^	7.723 × 10^−6^	1.592 × 10^−3^	5.712 × 10^−3^
F15	1.036 × 10^−3^	4.405 × 10^−1^	5.446 × 10^−2^	8.188 × 10^−5^	2.623 × 10^−1^	1.915 × 10^−1^	2.411 × 10^−4^
F16	1	1	1	1	1.734 × 10^−6^	1.734 × 10^−6^	6.25 × 10^−2^
F17	1.734 × 10^−6^	1.734 × 10^−6^	1.734 × 10^−6^	2.437 × 10^−4^	7.821 × 10^−6^	1.734 × 10^−6^	1.734 × 10^−6^
F18	7.626 × 10^−1^	2.186 × 10^−1^	1	3.75 × 10^−1^	1.734 × 10^−6^	1.734 × 10^−6^	1.47 × 10^−6^
F19	1	1	1	1	1.734 × 10^−6^	1.734 × 10^−6^	5 × 10^−1^
F20	2.492 × 10^−2^	6.875 × 10^−1^	2.148 × 10^−2^	9.271 × 10^−3^	1.494 × 10^−5^	1.891 × 10^−4^	6.088 × 10^−3^
F21	1.856 × 10^−2^	1.530 × 10^−4^	1.058 × 10^−3^	4.508 × 10^−2^	5.791 × 10^−2^	4.193 × 10^−2^	5.689 × 10^−2^
F22	1.114 × 10^−2^	8.596 × 10^−2^	1.397 × 10^−2^	8.365 × 10^−2^	1.975 × 10^−2^	6.732 × 10^−2^	1.556 × 10^−2^
F23	7.275 × 10^−3^	1.879 × 10^−2^	2.456 × 10^−3^	1.504 × 10^−1^	1.359 × 10^−2^	4.4913 × 10^−2^	7.139 × 10^−1^
+/=/−	16/4/3	15/2/6	18/3/2	16/4/3	17/3/3	19/0/4	18/0/5

**Table 7 biomimetics-08-00070-t007:** Experimental 1 parameter configurations.

Parameters	Values
Area of monitoring (*S*)	100 m × 100 m
Sensing radius (*R_s_*)	10 m
Number of sensors (*N*)	45
Number of iterations (*iteration*)	150

**Table 8 biomimetics-08-00070-t008:** Comparison of initial coverage and optimize coverage experimental results.

Sensors	Initial Coverage Ratio	Optimize Coverage Ratio
*N* = 45	79.13%	97.58%

**Table 9 biomimetics-08-00070-t009:** Comparison of experimental results from various algorithms.

Algorithm	Coverage Ratio	Coverage Efficiency
PSO	83.02%	58.75%
BES	91.32%	64.63%
WOA	94.62%	66.96%
HHO	96.48%	68.28%
WHO	92.06%	65.15%
IWHO	97.58%	69.08%

**Table 10 biomimetics-08-00070-t010:** Comparison of TABC and IWHO experimental results.

Method	Coverage Ratio
TABC [34]	96.07%
IWHO	97.58%

**Table 11 biomimetics-08-00070-t011:** Comparison of COOTCLCO and IWHO experimental results.

Method	Coverage Ratio
COOTCLCO [35]	96.99%
IWHO	99.17%

**Table 12 biomimetics-08-00070-t012:** Comparison of GWO-EH and IWHO experimental results.

Method	Coverage Ratio
GWO-EH [36]	83.64%
IWHO	97.78%

**Table 13 biomimetics-08-00070-t013:** Comparison of VFLGWO and IWHO experimental results.

Method	Coverage Ratio
VFLGWO [37]	94.52%
IWHO	99.48%

**Table 14 biomimetics-08-00070-t014:** Experimental 2 parameter configurations.

Parameters	Values
Area of monitoring (*S*)	100 m × 100 m
Sensing radius (*R_s_*_1_)	12 m
Sensing radius (*R_s_*_2_)	10 m
Communication radius (*R_s_*_1_)	24 m
Communication radius (*R_s_*_2_)	20 m
Number of sensor sensors (*N*_1,2_)	20
Number of iterations (*iteration*)	150

**Table 15 biomimetics-08-00070-t015:** Comparison of HWSN coverage optimization experimental results.

Sensors	Initial Coverage Ratio	Optimize Coverage Ratio
*N*_1,2_ = 20	81.43%	98.51%

**Table 16 biomimetics-08-00070-t016:** Comparison of HWSN connectivity optimization experimental results.

Sensors	Initial Connectivity Ratio	Optimize Connectivity Ratio
*N*_1,2_ = 20	16.03%	20.04%

**Table 17 biomimetics-08-00070-t017:** Comparison of HWSN obstacle coverage optimization experimental results.

Sensors	Initial Coverage Ratio	Optimize Coverage Ratio
*N*_1,2_ = 20	86.61%	97.79%

**Table 18 biomimetics-08-00070-t018:** Comparison of HWSN obstacle connectivity optimization experimental results.

Sensors	Initial Connectivity Ratio	Optimize Connectivity Ratio
*N*_1,2_ = 20	15.77%	17.44%

## Data Availability

Not applicable.

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
