# Peer review of "Coverage Optimization of Heterogeneous Wireless Sensor Network Based on Improved Wild Horse Optimizer"

_biomimetics, 2023, doi:10.3390/biomimetics8010070_

Round 1
Reviewer 1 Report
In this paper, a metaheuristic algorithm that integrates the wild horse optimizer algorithm and the golden sine cosine algorithm is developed. The proposed approach is tested to solve 23 standard benchmark problems and the heterogeneous wireless sensor network (HWSN) coverage problem.
This paper should be accepted subject to the following conditions:
1. The paper does not contain the proper references to existing literature. Previously developed hybrid metaheuristics related to the sine cosine algorithm and wild horse optimizer have to be mentioned in the paper. Some recently proposed algorithms are:
[REF 1] Li, Yancang and Yuan, Qiuyu and Han, Muxuan and Cui, Rong, Hybrid Multi-Strategy Improved Wild Horse Optimizer, Advanced Intelligent Systems, 4(10) 2200097, 2022.
[REF 2] Brajević, I.; Stanimirović, P.S.; Li, S.; Cao, X.; Khan, A.T.; Kazakovtsev, L.A. Hybrid Sine Cosine Algorithm for Solving Engineering Optimization Problems. Mathematics 2022, 10, 4555. https://doi.org/10.3390/math10234555
Also, on page 1, there is an error “1-Error! Reference source not found.” This should be corrected.
On page 2 it is written: “In 2022, Milovanović applied the WHO in multi-objective energy 50 management in microgrid [19]. ”
On the other hand the reference 19 is: Ali, M.; Kotb, H.; AboRas, M.K. Frequency regulation of hybrid multi-area power system using wild horse optimizer based new combined Fuzzy Fractional-Order PI and TID controllers. Alex. Eng. J. 2022, 61(12), 12187-12210,
and the coauthor Milovanović is not mentioned.
2. Brief literature review related to the standard sine cosine metaheuristic algorithm has to be mentioned in the paper.
3. A common recommendation for fair comparison amongst meta-heuristic algorithms is to compare algorithms based on an equal number of consumed fitness evaluations. Hence, in Section 6.6.1, number of consumed fitness evaluations for WHO, SSA, NGO, MA, PSO, COOT, GWO and the IWHO algorithms must be mentioned. Also, the specific control parameter settings of these metaheuristic algorithms must be explained in the paper.
4. In this paper, the statistical treatment of the data is not adequate. The appropriate statistical analysis must be done using the non-parametric statistical tests. Please see the paper:
[REF3] J. Derrac, S. García, D. Molina, F. Herrera, "A practical tutorial on the use of nonparametric statistical tests as a methodology for comparing evolutionary and swarm intelligence algorithms", Swarm and Evolutionary Computation, Vol. 1, No. 1, pp. 3-18, Mar 2011.
5. The advantages and the limitations of the proposed approach should be discussed in the paper.
Reviewer 2 Report
This paper first improves the WHO algorithm, and then applies it to optimizing the sensor coverage and connectivity in WSN. The experimental results show that the improved algorithm has better performance than other SI optimization algorithms. I recommend this paper for publication after several minor revisions.
1. Reference source is not found in Line 30.
2. Are ASFA and AFSA the same in Lines 56 and 57?
3. The lines 101 and 108 miss the “=”.
4. There are no spaces in front of “where”, such as Lines 222, 230 and 259.
5. The authors should supplement the convergence analysis of the proposed algorithm.
Round 2
Reviewer 1 Report
All my comments are properly addressed.